# Seeking Gastronomic, Healthy, and Social Experiences in Tuscan Agritourism Facilities

**Rosa Maria Fanelli**

Department of Economics, University of Molise, 86100 Campobasso, Italy; rfanelli@unimol.it;
Tel.: +39-874-404401

**Abstract:** There is a growing desire among tourists to improve their lifestyle and pass their vacations in areas with a strong local gastronomic heritage. Indeed, one of the most important factors that drives visitors to choose an agritourism facility for their vacation is the pull of good food and wine. This manuscript examines visitors' evaluations of their time spent in Tuscan agritourism facilities with particular focus on the cuisine, the health benefits, and the social experience. The analysis is based on a representative sample of 1886 reviews posted by visitors from all over of the world on the websites of 60 agritourism facilities that operate in Tuscan municipalities. By exploring visitor evaluations, which consist of an overall rating of their agritourism experience and separate ratings for the cuisine, the physical environment, and the service, this paper expands the scope of previous studies into food and drink management development as a way of attracting new visitors, by providing additional information on the distinctive characteristics of the local cuisine on offer. Results indicate that visitors—above all families—prefer an agritourism facility that offers authentic local cuisine and beverages and also offers them the possibility of spending time outdoors in destinations with a rich culinary heritage. Visibility of the main attractive attributes of agritourism facilities and the local cuisine through websites is needed in order to drive consumers in their vacation choices and to allow these structures to consolidate their place in the food and drink market.

**Keywords:** food vacations; motivational factors; Tobit regression; visitors' typologies; word-of-mouth

## 1. Introduction

In recent years, the level of demand from visitors for gastronomic, health, and social tourism has increased. Many tourists now seek food and drink experiences through a discovery of local culture, traditions and customs, and by building relationships with local and rural communities. Hand in hand with the desire for high quality local cuisine and thus the popularity of areas with a rich gastronomic heritage is the wish to lead a healthy lifestyle. For many travelers, food can be a motivation for travelling; it can represent a search for an extraordinary experience of life (Ellis et al. 2018). A 'food tourist' may be defined as a person who selects a travel destination as a result of the food experiences it can offer. Therefore, such a person gives great importance to the food, meal preparation, and food-related activities offered at a destination.

Food vacations are becoming a real player in the agritourism market; food and wine related services and events are attracting ever greater numbers of tourists each year and are key to our understanding of other cultures (Mason and Paggiaro 2012). Many tourists take the decision to visit a destination as a result of a wish to experience the local lifestyle and trying local food is an essential part of this. In accordance with Agrawal (2017), food should be considered as a good predictor of tourist satisfaction and represents a main reason why people travel to specific tourist destinations (Amira 2009). The characteristics of food reflect the authenticity of a destination's heritage and culture (Cankül and Demir 2018). Consuming local food not only increases a tourist's knowledge

about the local cuisine, but also motivates them to experience a destination's wider local culture (Carmen et al. 2017; Garibaldi and Pozzi 2018). Consequently, the regeneration of rural economies, the discovery of local identity, and the re-valuing of heritage and tradition can all flow from growing, processing, marketing, distributing, eating, and enjoying food and beverages. However, many aspects of a tourist's gastronomic experience, which include their search for a healthy lifestyle and their social experiences, appear to have been neglected in academic discourse. Indeed, research in this area is still in its infancy (Cohen and Avieli 2004; Correia et al. 2008; Henderson 2009; Kim et al. 2009; Kivela and Crotts 2005; Mak et al. 2012). Hence, identifying firstly the kinds of visitor to agritourism facilities and secondly, understanding the motivational factors that drive them to choose this type of vacation, are fundamental for both farm operators and destination marketers (Harrington 2005; Correia et al. 2008; Chang et al. 2011; Akdag et al. 2018). For this reason, it is very important to discover which motivational factors are dominant and how visitors describe their experiences of local cuisine and service in agritourism establishments.

The present study differs to others since it brings together different experiences of gastronomic tourism to assess the relationship between the following: The attributes of agritourism facilities; the typology of visitor; visitor motivation; and evaluations of the food and drink experience. In addition, it expands and reinforces our current understanding of the importance of word-of-mouth (WOM), focusing on the positive comments left by consumers online, which mainly describe their experience of the food and service on offer.

The remainder of the manuscript is structured as follows: In Section 2 a theoretical background and a literature review from which the research questions are derived is presented. Subsequently, in Section 3, the research methodology, which includes the data collection methods, is presented. In Section 4, the data analysis, the results, and a detailed discussion of the key findings and their implications are provided. Finally, in Section 5, some summary conclusions and the limitations of the study are reported.

## 2. Theoretical Background and Literature Review

In the previous literature, the motivational factors that drive tourists in their choice of vacation destination and agritourism structure have been approached by authors from different perspectives. In this research, the aim is to focus attention on three aspects extrapolated from a review of the theoretical background and other empirical studies. The first aspect chosen is the role gastronomy plays in people's lives and the motivational factors that drive different typologies of visitor to destinations with strong culinary traditions. The second aspect concerns the importance of an informal channel (WOM) in determining the choice of an agritourism vacation. Finally, the third aspect regards the importance of the website as a tool for providing consumer information and for consolidating the presence of farm operators in the food and drink market.

### 2.1. The Role of Food Consumption in the Search for a Healthy Lifestyle and the Motivational Factors That Drive Visitors to Choose Gastronomic and Social Experiences

Food plays a multi-functional connecting role in society and influences people's lifestyle, health, and habits (Morgan 2010). The attractions of food are so persistent, they shape our lives in a variety of ways and have implications for all aspects of our life, especially for our social life. Eating is a pivot around which our social lives revolve, and feeding ourselves and others well is an essential part of socializing. The pangs of hunger are not only a reason to seek pleasure; they are a reason to seek friendship. This is the wisdom to which Epicurus alludes and foodies and wine lovers embody (Allhoff and Monroe 2009). Furthermore, in 2014, with the approval of a motion for the European Parliament Resolution on "European gastronomic heritage", the Committee on Culture and Education of the European Parliament recognized the importance of food and gastronomy as artistic and cultural expression and fundamental pillars of family and social relationships (European Parliament 2014).

Another important event for an analysis of the role of food in addressing nutrition, food production, management and distribution, as well as global and regional food governance, was the expo 2015, "*Feeding the Planet, Energy for Life*". This event sought to foster an international dialogue concerning nutrition and natural resources. Food is not only vital for survival and well-being, it is also a key part of all cultures, a major element of intangible heritage, and an increasingly important attraction for visitors. The linkages between food and tourism also provide a platform for local economic and agriculture development and local products, which can be strengthened by the use of food experiences for branding and marketing destinations. For this reason, in accordance with Levitt et al. (2017), it is very important to identify the different typologies of visitor, each with their own interests, who seek gastronomic and social experiences. Such a segmentation will help farm operators to customize their different food services to different types of visitor.

Regarding the different motivational factors that drive visitors to choose gastronomy and experiences that promote a healthy lifestyle, some studies (Civitello 2008; Leong et al. 2017) have argued that gastronomy can be used as a way of building social relationships with local communities. Furthermore, tourists who have such experiences have the possibility to acquire new knowledge about local gastronomic culture (Herrera 2012; Stanley and Stanley 2015). In addition, other authors (Ryu and Jang 2006; Mak et al. 2012) have argued that tourists who have previous experience of local gastronomy are likely to want to repeat it. Further studies have shown how local cuisine and wine are among the attractions most favored by tourists in Italy (Baloglu and Mangaloglu 2001; Brown and Getz 2005). An empirical study recently conducted in this field was carried out by Pérez et al. (2017) and analyzed attitudes concerning the local cuisine among foreign travelers who visited the city of Quito. Building on this empirical research, the following research questions were set:

**RQ₁:** *What types of visitor seek gastronomic, health and social experiences in Tuscan agritourism facilities?*

**RQ₂:** *What are the motivational factors that drive visitors to choose Tuscan agritourism facilities?*

*2.2. The Role of Online Reviews in Determining Visitor Destination Choices*

Tourists can obtain information about which gastronomic tourism destinations to choose for their vacations through both formal and informal channels (Mathieson and Wall 1982). Informal channels can relate to online WOM (or electronic word-of-mouth, or 'word-of-mouse') and in this case is represented by posts from parents, friends, families, other travelers, and bloggers who have been to a particular destination. As is well known, customers in the tourism sector are diverse and have different expectations and therefore can be difficult to satisfy. Consequently, the advent of online reviews has provided a plethora of information that could help satisfy visitor needs and has transformed consumer behavior in terms of searching for and sharing information. Indeed, in the literature there are indications that online consumer reviews have a significant influence on travel information searches and product sales, and play an increasing role in consumer purchase decisions (Duan et al. 2008; Litvin et al. 2004; Mauri and Minazzi 2013; Xiang and Gretzel 2010). One of the reasons behind the growth in popularity of online review websites is that they empower consumers by allowing them to share their experiences with others online, instead of having to meet them face-to-face (Sparks and Browning 2010). In addition, online review websites allow consumers to search for detailed and reliable information by sharing their consumption experiences (Dellarocas 2003; Yoo and Gretzel 2008; Fanelli and Nocera 2018) with the aim of reducing their level of perceived uncertainty (Ye et al. 2009) and of promoting local food and gastronomy tourism (Lim et al. 2019). Indeed, visitor satisfaction is important for successful destination marketing as it influences the selection of the destination, the consumption of products and services, publicity WOM, and the decision to return (Kozak and Rimmington 2000). In some studies (Chi and Qu 2008), the satisfaction of tourists has been measured by gauging general satisfaction with attributes (e.g., attractions, accommodation, accessibility, amenities, and activities) and has been transmitted through the written word. This method has the advantage of

allowing consumers to seek information at their own pace (Sun et al. 2006). The research questions on the role of online reviews in determining the destination choices of visitors derived mainly from the latest research.

**RQ₃:** *How do visitors judge online agritourism facilities, especially the food services?*

**RQ₄:** *What positive online comments do visitors use to describe their gastronomic and social experiences in Tuscan agritourism facilities?*

*2.3. The Role of Websites in Determining the Visibility of Farm Operators in the Food and Drinking Market*

Nowadays, the virtual world (Internet) plays a dominant and increasingly important role in driving tourists to choose their vacation destination. For this reason, in the agritourism sector, websites can have an important role in promoting products and services in specific areas (Cafiero et al. 2019) and can represent a rich and useful resource not only for customers, but also for marketers evaluating product and service quality (Decker and Trusov 2010). Fortunately, in recent years, the owners of many agritourism structures have understood that websites, often made in an amateur manner and at the lowest possible cost (Król 2019), are an important tool for spreading information and for embracing local cuisine and food culture as a vehicle for destination marketing (Mohamed et al. 2019). Nevertheless, several authors have found that the quality of website design is very important not only for attracting real and potential consumers and affecting their purchasing intentions (Lee and Lin 2005; San Martín and Herrero 2012), but also for tourism development and competition (Beldona and Cai 2006; Kim et al. 2007). In practical terms, web pages represent a modern marketing tool for the operators of agritourism facilities that can attract new customers.

Considering these functions, it is necessary that the information that the home page (website) contains is able to communicate the distinctive characteristics of the agritourism structure, the services and activities on offer, the local gastronomic identity, and who the farm operators are. Based on these considerations, the following question was formulated:

**RQ₅:** *How can websites help farm operators implement a role for themselves in the food and drink market?*

**3. Methodology and Research Design**

The literature background outlined above provided the impetus for this research. The paper therefore seeks to present an empirical insight into the most important motivations that drive visitors to consume local and traditional products in agritourism structures and seek a new life philosophy. A clear example of gastronomic heritage is found in Tuscany, where the combination of a rich, attractive culture and a history of culinary specialties makes the region immensely popular (Lemmi and Tangheroni 2015). This study regards a selected sample of agritourism facilities associated with the Agriturismo.it organization (Agriturismo.it 2018). The data were extrapolated from individual agritourism structure websites. As has been reported, the website plays an important role in determining consumption choice, in bringing potential benefits, and in reaching new customers. Thirty-three variables were proposed and reasons for their inclusion offered. Data were analyzed using descriptive statistics, the Tobit regression technique, and qualitative analysis.

In the first step, with the assistance of descriptive statistics, the most important attributes of the Tuscan agritourism facilities, the principal typologies of visitor, and the most relevant motivational factors that guide visitor destination choice, were identified.

In the second step, the Tobit regression technique for analysis was adopted to estimate how much of the variation in visitor evaluation of restaurants could be explained by independent variables. In contrast to other techniques, this model, which was devised by Tobin (1958), makes it possible to use all variables, both those with limiting values (usually 0) and those with higher values, in order to estimate a regression line.

The dependent variable was identified as the evaluation of the restaurant (EVALREST). This is a discrete variable, with a variation range between 0 and 10; $\alpha$ = the intercept of the regression equation; and $\beta k$ = coefficients of independent variables, where $k$ = 1, 2, 3, … 33 and $\varepsilon$ = error term. It was hypothesized that this evaluation would be influenced by the three following blocks of independent variables:

### 3.1. Attributes of Agritourism Facilities

Altitude (location of the agritourism facility, expressed in meters above sea level) = discrete variable.

Evaluation (score) of all the characteristics of all structures = discrete variable. This variable has a range between 0 and 10. It was hypothesized that visitor evaluation of the gastronomic experience would be positively related to all the agritourism facility attributes (surrounding area, activities, facilities, traditions, cultural heritage, and so on). Visitor frequency (Nr/Ny). This variable was expected to have a positive relationship with the dependent variable because it was hypothesized that a high number of reviews posted online by visitors and therefore the sharing of their experience would be a positive sign for the reputation of food services.

### 3.2. Typology of Visitor

It was hypothesized that some typologies of visitor would be more interested in this type of vacation and influenced in a more positive way by the dependent variable, since judgements depend on the level of visitor satisfaction with the agritourism structures.

### 3.3. Motivational Factors

The motivational factors that drive visitors to choose this type of vacation can influence in different ways both their gastronomic and social experiences in agritourism structures and their search for a healthier way of living while they are there.

The testable model for Tobit regression was thus:

$$EVALREST\ i,t = \alpha + \beta j\Sigma\ (Attributes\ of\ agritourism\ facilities)\ i,t + \beta m\Sigma\ (Typology\ of\ visitors)\ i,t + \beta n\Sigma\ (Motivational\ factors)\ i,t + \varepsilon i,t$$

In the last step, for the qualitative analysis, the large variety of online reviews and the repetition of particular comments concerning visitor opinions of agritourism facilities made it possible to describe visitors' positive experiences of seeking a philosophy of life through the consumption of local foods and beverages. This information could aid farm operators in their implementation of a quality website and quality food services.

The results of the study in relation to the above five research questions are described below.

## 4. Results and Discussion

The first question of this study aimed to identify and analyze the different typologies of visitor that spend a period of their vacations in Tuscan agritourism facilities. The choice to analyze agritourism facilities operating in Tuscan municipalities was made in view of the fact that this Italian region offers an interesting example of the transformative power of tourism and the commodification of rural areas, with a particular focus on culinary tourism. Indeed, over the past 30 years Tuscany has become a leader in the agritourism sector, which has helped to shape the region's iconic tourism identity (National Tourism Agency 2012). ISTAT data from 2013 indicate that the percentage of *agriturismi* located in Tuscany made up 19.7% of the total at the national level (4108 of 20,897 in Italy) (Istat 2013). Many farms have undertaken the transition towards tourism accommodation and today, *agriturismi* account for over 60% of beds in many rural municipalities. Furthermore, in many Tuscan municipalities, like in those of other Italian regions, especially in rural areas, the agritourism sector is now one of the main sources of employment, where traditional medium and small-scale farming dominates the scene

(Fanelli 2018). The experiences of the agritourism offer in Tuscany presents a useful model for those wishing to introduce heritage tourism (and specifically culinary tourism) in other countries of the world in an attempt to address rural issues linked to modernity.

A summary of the attributes of agritourism facilities, the typologies of visitor, and the motivational factors that drive visitors to spend a period of their vacations, usually a short weekend break, in Tuscan agritourism facilities is shown in Table 1. The agritourism facilities are located in hilly municipalities between 3 and 814 m (m) above sea level (ASL) with a mean of 289.45 m ASL. On average, the visitors are predominantly made up of families with children (12.23), couples (8.75), anonymous (7.35), and groups of friends (1.28). The values of coefficient of variation show that there is heterogeneity among the agritourism facilities in relation to some typologies of visitor (such as business groups, disabled people, motorcyclists, and students). A higher visiting rate for families with children (39%) is a trend that has also been observed by other authors (Indrová et al. 2008). Indeed, the most popular form of tourism for families with children is rural tourism. This may be due to the fact that, especially during the summer season, children are normally provided with an outdoor playground and swimming pool. Furthermore, children and their families can enjoy themselves and their opinion of what constitutes a healthy lifestyle can be reinforced. Furthermore, it is predicted that tourism for families with children will grow more than any other form of tourism, since a family vacation offers an opportunity for family members to be united and spend time together (Niemczyk 2015). Another larger share of visitors (28%) is represented by couples who spend a period of their vacations in agritourism structures, often as part of their honeymoon. These visitors seek contact with nature and the opportunity to taste good food and wine in a romantic setting.

**Table 1.** Descriptive statistics.

| Attributes and Visitors' Evaluation of Agritourism Facilities | Mean | Std. Dev | Min | Max | Coeff. of Var. |
|---|---|---|---|---|---|
| Altitude | 289.45 | 199.07 | 3 | 814 | 0.69 |
| Evaluation of all structure characteristics | 9.57 | 0.57 | 6.3 | 10 | 0.06 |
| Evaluation of restaurant | 9.53 | 0.89 | 4.3 | 10 | 0.09 |
| Visitor frequency | 4.10 | 3.25 | 1 | 15.56 | 0.79 |
| **Typology of Visitors** | **Mean** | **Std. Dev** | **Min** | **Max** | **Coeff. of Var.** |
| Couples | 8.75 | 10.39 | 0 | 43 | 1.19 |
| Singles | 0.42 | 0.67 | 0 | 3 | 1.60 |
| Families with children | 12.23 | 16.72 | 0 | 92 | 1.37 |
| Groups of friends | 1.28 | 1.54 | 0 | 5 | 1.20 |
| Independent travelers | 0.65 | 1.10 | 0 | 4 | 1.69 |
| Elderly visitors | 0.43 | 0.72 | 0 | 3 | 1.67 |
| Students | 0.17 | 0.56 | 0 | 3 | 3.29 |
| Disabled people | 0.05 | 0.22 | 0 | 1 | 4.40 |
| Business groups | 0.03 | 0.18 | 0 | 1 | 6.00 |
| Motorcyclists | 0.07 | 0.25 | 0 | 1 | 3.57 |
| Anonymous | 7.35 | 15.60 | 0 | 93 | 2.12 |
| **Motivational Factors** | **Mean** | **Std. Dev** | **Min** | **Max** | **Coeff. of Var.** |
| Fun | 1.25 | 1.50 | 0 | 6 | 1.20 |
| Organic | 0.30 | 0.91 | 0 | 6 | 3.03 |
| Relaxing | 6.57 | 8.14 | 0 | 43 | 1.24 |
| Artistic/cultural | 2.63 | 4.27 | 0 | 21 | 1.62 |
| Romantic | 1.57 | 2.31 | 0 | 11 | 1.47 |
| Short stays | 1.80 | 2.50 | 0 | 13 | 1.39 |
| Long stays | 0.10 | 0.35 | 0 | 2 | 3.50 |
| Children welcome | 5.82 | 8.86 | 0 | 46 | 1.52 |
| Seaside | 0.38 | 1.30 | 0 | 7 | 3.42 |
| With specific courses | 0.05 | 0.29 | 0 | 2 | 5.80 |
| Honeymoons | 0.17 | 0.46 | 0 | 2 | 2.71 |

**Table 1.** *Cont.*

| Motivational Factors | Mean | Std. Dev | Min | Max | Coeff. of Var. |
|---|---|---|---|---|---|
| Great food/wine | 1.40 | 2.27 | 0 | 10 | 1.62 |
| Business trips | 0.12 | 0.37 | 0 | 2 | 3.08 |
| Educational activities | 0.08 | 0.28 | 0 | 1 | 3.50 |
| Open air | 0.48 | 0.97 | 0 | 5 | 2.02 |
| Sports | 0.22 | 0.45 | 0 | 2 | 2.05 |
| Spa baths | 0.17 | 0.74 | 0 | 5 | 4.35 |
| Adventure | 0.17 | 0.42 | 0 | 2 | 2.47 |
| Not specified | 8.17 | 16.08 | 0 | 95 | 1.97 |

The second research question sought to identify the motivational factors that drive visitors to choose an agritourism facility destination. The principal motivations that drive visitors to spend a period of their vacation in Tuscan agritourism facilities, in accordance with Ting et al. (2017), are varied. On average, the most important motivational factors that drive the different typologies of visitor, excluding those who did not specify their motivation, can be summarized thus: Relaxation (6.57), children welcome (5.82), artistic culture (2.63), short stays (1.80), great food/wine (1.40), and fun (1.25). Other factors given lesser importance are reported in Table 1. The findings are in accordance with other studies (Chinnici et al. 2013 Platania and Privitera 2006). During their stay, visitors have the opportunity to get closer to nature (open air), learn about and appreciate local products (great food/wine), and learn about rural culture (educational activities, specific courses, and so on). The search for open spaces and an experience of imagined rurality drives visitors to choose this form of tourism. Proposals made by farmers are often limited to activities that can be enjoyed in one day without an overnight stay. This is true, for example, in the case of educational farms, which have witnessed a growth in visitor interest. Research conducted by ISMEA (2009) provides information on tourist expectations. On the one hand, tourists consider the location of the agritourism structure important, imagining it to be immersed in the quiet countryside. On the other hand, they consider the presence of the agricultural entrepreneur inside the farm fundamental to ensure a welcome that is familiar and absolutely different from that of a hotel structure. There is a clear will to escape the stress and freneticism that characterizes working life, where individuals are often forced to spend most of their time in enclosed spaces which are often chaotic. Rurality, in the imagination of these individuals therefore is synonymous with calm, time, authenticity, and open air. Following this logic, agritourism structures are not expected to be too big; and they are expected to have a limited number of rooms, which are not necessarily luxurious but make use of local and natural products. The catering service is not expected to use processed or frozen food but provide typical dishes of the area. The farmer is required to be able and willing to provide information about the surrounding area and its practices in a relaxed way, dissimilar to the frenetic communications of city life.

At this point of the analysis, it is very important to understand visitor judgements of the attributes of the agritourism facilities, especially those relating to food services. The judgment scores given by visitors concerning all the characteristics of the agritourism facilities and those relating to the restaurant are very high (the scores are 9.57 and 9.53, respectively). In addition, there is, with a low value of the coefficient of variation, sufficient homogeneity among the structures of the sample analyzed. Another important indicator is visitor frequency, calculated as the ratio between the number of reviews and the number of years during which the evaluations were made. In this case, the value of the coefficient of variation is high and highlights that in this niche of the Italian agritourism sector, agritourism facilities with a long presence online coexist with those that have only more recently come to appreciate the importance of the website as a showcase for potential visitors.

In the second step, the Tobit regression technique was adopted to estimate how much of the variation amongst the visitors' evaluations of restaurants was explained by the independent variables. As indicated in Table 2, the pseudo R-squared value indicated that 68% of this variation could be explained by the independent variables. In more detail, the visitor evaluation of food and drink was

significantly determined by all the characteristics of the agritourism structure (i.e., accommodation and activities and facilities present in the countryside where the structures are located). In contrast, the evaluation of restaurants increased by about 11% for every additional review added to the website of an agritourism facility. The visitor frequency is used as a proxy to measure the consolidated presence of the structure's personal website and its association with the larger Agriturismo.it organization. It is believed that the association of a structure to the Agriturismo.it organization gives more visibility to its characteristics and to the evaluations of other visitors, facilitating the choice of a particular destination. Furthermore, the consolidated presence online is significantly associated with a high rate of visitor frequency. In accordance with Wan (2002), the information that potential customers collect through the Internet (website) is very important to make an informed choice, since it gives visitors the opportunity to compare different possibilities. Furthermore, the results are consistent with those of Kim et al. (2007), who found that in the last decade, the website has become an important distribution channel and a tool, which is especially important for competition and agritourism development.

**Table 2.** Tobit regression estimates of determinates of the evaluation of restaurant attributes.

| Evaluation of Food and Drink (Restaurant Score) | Coefficient | Std. Err. | t | $P > t$ | [95% Conf. Interval] | |
|---|---|---|---|---|---|---|
| Constant | −3.318 | 1.142 | −2.91 | 0.007 | −5.657 | −0.979 |
| Altitude | 0.000 | 0.000 | 0.39 | 0.696 | −0.001 | 0.001 |
| Evaluation (score) of all structure characteristics | 1.326 *** | 0.122 | 10.85 | 0.000 | 1.076 | 1.576 |
| Nr/Ny | 0.133 ** | 0.097 | 1.37 | 0.182 | −0.066 | 0.332 |
| Couples | −0.064 * | 0.049 | −1.29 | 0.208 | −0.165 | 0.038 |
| Singles | −0.189 ** | 0.154 | −1.23 | 0.229 | −0.505 | 0.126 |
| Families with children | −0.104 ** | 0.035 | −2.96 | 0.006 | −0.177 | −0.032 |
| Groups of friends | −0.0425 * | 0.075 | −0.57 | 0.574 | −0.195 | 0.110 |
| Independent travelers | 0.128 ** | 0.107 | 1.19 | 0.243 | −0.092 | 0.347 |
| Elderly people | 0.209 ** | 0.132 | 1.58 | 0.125 | −0.062 | 0.480 |
| Students | 0.189 ** | 0.267 | 0.71 | 0.486 | −0.359 | 0.736 |
| Disabled people | −0.334 * | 0.296 | −1.13 | 0.269 | −0.941 | 0.273 |
| Business groups | 0.714 ** | 0.594 | 1.20 | 0.239 | −0.502 | 1.930 |
| Motorcyclists | −0.900 ** | 0.569 | −1.58 | 0.125 | −2.066 | 0.2663 |
| Anonymous | 0.000 | 0.009 | 0.05 | 0.960 | −0.017 | 0.018 |
| Fun | 0.048 * | 0.091 | 0.53 | 0.601 | −0.138 | 0.234 |
| Organic | 0.098 * | 0.126 | 0.78 | 0.440 | −0.159 | 0.356 |
| Relaxing | 0.026 * | 0.042 | 0.63 | 0.535 | −0.059 | 0.111 |
| Artistic / cultural | 0.065 * | 0.0421 | 1.56 | 0.131 | −0.021 | 0.151 |
| Romantic | 0.0259 * | 0.055 | 0.47 | 0.644 | −0.087 | 0.139 |
| Short stays | 0.092 * | 0.085 | 1.08 | 0.289 | −0.082 | 0.266 |
| Long stays | −0.419 ** | 0.225 | −1.86 | 0.073 | −0.880 | 0.042 |
| Children welcome | 0.101 ** | 0.042 | 2.40 | 0.023 | 0.015 | 0.187 |
| Seaside | −0.016 * | 0.107 | −0.15 | 0.883 | −0.235 | 0.204 |
| With specific courses | 0.666 ** | 0.499 | 1.33 | 0.193 | −0.357 | 1.688 |
| Honeymoons | −0.093 * | 0.242 | −0.38 | 0.705 | −0.587 | 0.402 |
| Great food/wine | 0.054 * | 0.076 | 0.71 | 0.484 | −0.102 | 0.209 |
| Business trips | −0.178 ** | 0.436 | −0.41 | 0.686 | −1.070 | 0.715 |
| Educational activities | −0.626 ** | 0.418 | −1.50 | 0.145 | −1.482 | 0.230 |
| Open air | 0.364 ** | 0.160 | 2.28 | 0.031 | 0.037 | 0.692 |
| Sports | −0.026 * | 0.218 | −0.12 | 0.906 | −0.473 | 0.421 |
| Spa baths | −0.051 * | 0.133 | −0.38 | 0.706 | −0.324 | 0.222 |
| Adventure | 0.028 * | 0.208 | 0.13 | 0.894 | −0.398 | 0.454 |
| Not specified | 0 | | | (omitted) | | |
| Number of observations | 60 | | | | | |
| F (32, 27) | 4.20 | | | | | |
| Prob > F | 0.000 | | | | | |
| Pseudo R2 | 0.683 | | | | | |
| Log Pseudo likelihood | −25.207 | | | | | |

*** $p < 0.01$, ** $p < 0.05$ and * $p < 0.10$.

Tourists, therefore, choose facilities mainly on the basis of the word of friends. For this reason, the presence of appropriate information about the structure and location but also an accessible and well-structured website are variables of strategic importance that can influence destination choice. Furthermore, visitor reviews and comments have an impact on the reputation of organizations. However, the results show that the agritourism facilities with a more recent presence online have a lower reputation and are less visited.

Regarding the typologies of visitor, the number of families with children influenced in a stronger manner the evaluation of food and wine: The score decreased by about 3% for every 0.10 decrement in families with children. This is an important driver; families with children choose agritourism structures mainly on the basis of the judgments given by previous visitors. Parents are usually very attentive to the quality of the food that their children will have to eat.

Finally, among the motivational factors that drive guests to spend a period of their vacation in agritourism structures, those that affect the evaluation of the restaurant the most are the following: Children welcome, the possibility of dining outdoors, and a long stay in an agritourism facility.

The findings suggest that when people can choose agritourism structure, they prefer those that have activities and facilities for children that are located in an unspoiled natural environment and where they can stay for a long time and thus experience the pleasure of rural life and taste high-quality local food and wine.

The most frequently occurring positive comments in the sampled reviews are listed in Table 3. Only positive comments used more than 20 times were included, as these were therefore likely to have been found in at least 20% of the sampled reviews. The frequency of the words "good," "excellent," "fresh," "excellent company," and "local products" can be seen to illustrate the strength of opinion found in the reviews.

These positive views were to be expected. As previous studies have shown, the majority of online reviews are positive and trustworthy (Masłowska et al. 2017; Anaya-Sánchez et al. 2019). This is consistent with the argument that users are more likely to post a review if they feel strongly about the subject. Comments by visitors, however, depend primarily on the consumer experience, and only after experiencing the service can the consumer make a judgment.

**Table 3.** Frequently occurring positive comments by visitors.

| Expressions | Frequency | % |
|---|---|---|
| Fresh, regional and typical Tuscan cuisine | 210 | 11.13 |
| Delicious cuisine and a real pleasure to have dinner with the whole family | 125 | 6.63 |
| The food is superb, biodynamic, organic and fresh from owner's farm | 100 | 5.30 |
| Much appreciated dinner with guests (from all over the world) | 85 | 4.51 |
| Delicious vegetarian dinners, "simple" and healthy | 84 | 4.45 |
| Atmosphere is really pleasant in contact with nature | 75 | 3.98 |
| The food is delicious and we learned to make pasta with the owners | 72 | 3.82 |
| Excellent kitchen facilities | 70 | 3.71 |
| Sample authentic cuisine | 70 | 3.71 |
| Delicious, typical and regional dinner menus | 67 | 3.55 |
| Delicious breakfasts and dinners prepared with care and with local products | 65 | 3.45 |
| We ate around the same table the delicious dishes in a harmonious and joyful atmosphere | 64 | 3.39 |
| Delicacies prepared with passion and served with grace and courtesy | 60 | 3.18 |
| The extraordinary meals are tasty and generous | 58 | 3.08 |
| Great food, good cook | 55 | 2.92 |
| Excellent cuisine, excellent company | 55 | 2.92 |
| Cooked great dishes were shared by all guests in the garden in such a lovely evening | 54 | 2.86 |
| The cuisine is excellent, prepared with quality and authenticity, accompanied by fine wine | 52 | 2.76 |
| The cuisine is sublime, all handmade with quality products | 51 | 2.70 |
| You eat divinely | 43 | 2.28 |
| The kitchen is simple and well taken care of all based on local produce and homemade. | 42 | 2.23 |

**Table 3.** *Cont.*

| Expressions | Frequency | % |
|---|---|---|
| Good | 40 | 2.12 |
| The food is delicious and the people very helpful | 40 | 2.12 |
| Intimate and convivial environment of breakfasts and dinners, always prepared with gusto and passion | 36 | 1.91 |
| Exceptional service | 35 | 1.86 |
| The meals are prepared with love and you can taste it! | 31 | 1.64 |
| More than good—at least for my taste. | 29 | 1.54 |
| Excellent cuisine, with dishes prepared using the products of family farming. | 27 | 1.43 |
| The authentic and local cuisine | 25 | 1.33 |
| Tasty food tasting in company | 23 | 1.22 |
| Food for the eyes and soul! | 22 | 1.17 |
| Very friendly staff | 21 | 1.11 |
| Total | 1886 | 100 |

Finally, in relation to last question: *How can websites help farm operators implement a role for themselves in the food and drink market?* The findings of the study, in accordance with Saxena and Ilbery (2010), suggest that the development prospects for the agritourism appear to be linked to the ability of farmers to offer authentic, fresh products, which are typical of their location. Visitors are seeking new experiences and are keen to reconnect with the cultural roots of food in the places from where the ingredients and gastronomy originate (Sidali et al. 2015). Indeed, some studies have indicated that local cuisine, including wine, is among the attractions most favored by tourists in Italy (Baloglu and Mangaloglu 2001; Brown and Getz 2005). The agritourism experience gives visitors the possibility to spend their vacation in a territory where a specific food or wine is prepared or produced by the local community, with local and organic raw materials. Indeed, in Tuscan agritourism facilities, the guests "live" on the farm, consume its products, and participate actively in its activities. In this field, Tuscany represents a model to which many other Italian regions and world countries are looking with increasing interest.

Furthermore, the results of the study, in line with the data of the Osservatorio Agrie Tour (2016), highlight that the 1886 visitors to Tuscan agritourism structures chose to visit in order to experience all the peculiarities of this kind of vacation: Nature, food and wine, relaxation, and activities inside and outside the farm.

Finally, the qualitative analysis shows that reviews of an agritourism's facilities center around a number of key nouns (nature/landscape, cuisine, history/art, relaxation, hospitality) and adjectives (genuine, traditional, authentic, fresh, organic). Therefore, one way to increase the attractiveness of products and services to potential future visitors, in accordance with Oliveira and Casais (2019), could be to publish pictures of food and other physical evidence from restaurants taken by visitors. On the basis of visitor experiences, farmer operators have the opportunity to decide which aspects of their structures they should enhance over others.

Moreover, the analysis shows that both the professionalism and the friendliness of staff, which leads to relationships being established, are a distinctive feature of agritourism when compared to other more impersonal tourism enterprises. The results corroborate the findings of research conducted by Sanchez-Cañizares and Castillo-Canalejo (2015) on a representative sample of 392 tourists in the cities of Ljubjlana and Cordoba. They found that that human capital formation is an important determinant for the enhancement of the international visibility of emerging culinary tourism.

## 5. Conclusions

The goal of this research was to examine the motivational factors that drive visitors to seek a gastronomic, healthy, and social experience within agritourism facilities and to analyze the online evaluation of the attributes of these structures, especially in relation to food services. In this field, scientific research is still in its infancy and often focuses merely on analyzing the importance of online

reviews for the development of the agritourism sector. Indeed, only a few studies have provided empirical analysis (Cohen and Avieli 2004; Correia et al. 2008; Henderson 2009; Kim et al. 2009; Kivela and Crotts 2005; Mak et al. 2012). This paper builds on this analysis by using the websites of Tuscan agritourism facilities associated with the Agritourismo.it organization to define—through an analysis of the characteristics of agritourism structures and online reviews—the typologies of visitor and visitor motivation. In addition, the study extends previous research by assessing the impact of online reviews on customer evaluations of food and drink quality and service delivery. The research results demonstrate that visitors, above all families, prefer an agritourism facility that offers good food and drink and that gives them the possibility to spend time with their children outdoors, in locations with a strong gastronomic cultural heritage. Indeed, the mean recorded for each typology of visitor suggests that families with children, who, as reported in the literature, were expected to visit agritourism facilities more frequently, dominated the sample of visitors. In addition, the analysis shows that families with children are willing to explore outdoor recreation possibilities such as sport and leisure activities (e.g., hiking, bird watching, hunting, fishing, horse riding, diving, golf, tennis, and shopping) and to take part in specialist courses (e.g., cookery classes, educational activities) on offer. Families also believe that such activities are very important for their health and well-being. The results confirm the data of the Osservatorio Agrie Tour (2016), which showed that families choose farms to relax and taste the local cuisine but also to visit natural or historical attractions in the surrounding area and take part in activities on the farm. Families are particularly receptive to farms that offer a family atmosphere and of course, spaces and activities dedicated to children. Furthermore, the high average score (9.53) given to agritourism restaurants by visitors suggests that the restaurant is one of the most important reasons for visiting such structures. In the present study, scores given to restaurant quality were higher when the reviews were predominantly positive, a finding that emphasizes the persuasive role played by positive consumer feedback on the perceptions of future customers (Donovan and Jalleh 1999). Indeed, the experience of the restaurant seems to leave the most lasting impression on visitors to Tuscan agritourism facilities.

The findings of this study could serve as an impetus for further research into online promotions in the agritourism sector. Indeed, it could be useful to explore visitor views of food and wine, aspects which can be seen as integral and indispensable to agritourism vacations. A philosophy of life that promotes local cuisine and the cultural heritage of an area is attractive to both Italian tourists and to those from further afield. The worldwide renown of the Italian culinary tradition is a particular pull for tourists from outside of Italy.

This study has some limitations that may offer directions for future research. First, the analysis is restricted to the evaluation of the gastronomic side of agritourism facilities operating in Tuscany. Future research could take the current study further by carrying out a review of international websites, which could improve external validity and examine the differences in heterogeneous areas. The conclusion that emerges from this paper is that online WOM cannot be ignored. It will continue to have a more important role than traditional WOM in the promotion of gastronomy, well-being and social relations and can serve as leverage to motivate visitors. The motivation factors and the comments left by visitors describing their experiences could help farm operators to better understand the wants and needs of their customers. It is very important to provide an appropriate approach for farm operators associated with the Agriturismo.it organization and thereby assist them in developing strategies that would best develop this market.

**Funding:** The funders had no role in the design of the study; in the collection, analysis, or interpretation of data; in the writing of the manuscript; or in the decision to publish the results.

**Acknowledgments:** The author of this paper wishes to thank the anonymous reviewers for the interesting suggestions they made for the revision of this paper.

**Conflicts of Interest:** The author declares no conflict of interest.

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
