# Peer review of "Seeking Gastronomic, Healthy, and Social Experiences in Tuscan Agritourism Facilities"

_socsci, doi:10.3390/socsci9010002_

Round 1
Reviewer 1 Report
Suggestions for improvements:
THE INTRODUCTION
The first part of the paper should be restructured. I propose a shorter Introduction, and a theoretical section.
Information about the empirical context (Toscana) could be placed at the beginning of the results as a Background information section.
Important: the theoretical basis of the study should go deeper in some concepts. For example: it is unclear what it is meant with philosophy: is it the same than lifestyle (not really, so please clarify the use of this term). Please, make sure that your theoretical approach has some aspect of ORIGINALITY in comparison with the numerous studies about food and wine in rural tourism. I'm wondering whether the focus of this study is on the promotion of agriturismo, as this is what it is commented at the end of the paper. Please, clarify and go deeper in the theoretical approach/perspective.
Pay attention to the order of the various topics within this section. As it is now, the text "jumps" from a topic to another and then back to the first topic and so on. A possible order for the Introduction might be: food & drink (importance, identity issues...), food & drink in rural tourism, presentation of ONE main RESEARCH QUESTION, + final paragraph presenting the structure and content of the following sections. Similarly, the theory chappter could begin with some considerations about already available studies about food tourism in rural areas, and then highlights what YOUR study will contribute with.
METHOD
The choice of Toscana as the emprical setting of the study should be explained.
More info about the sources of data, in particular the reviews: it seems that the reviews are from the agriturismo's webpage (not very reliable, as the agriturirsmo might delete bad reviews), or other online platforms? This is an aspect that should be explained. The possibility of faked reviews should also be commented. A table including such ino, as well as the number of reviews could be useful.
Please insert a CONCLUSION section that summarizes the MAIN findings, and highlights the theoretical and practical contribution of your study.
Avoid very short paragraphs (1 sentence)
Author Response
Reviewer 1
THE INTRODUCTION
The first part of the paper should be restructured. I propose a shorter Introduction, and a theoretical section.
Response:
In the new version of the paper, I have proposed a shorter Introduction (paragraph 1) Theoretical background and literature review, instead, are reported in the paragraph 2 (see the new version of the paper from pag. 2 to pag 4).
Information about the empirical context (Toscana) could be placed at the beginning of the results as a Background information section.
Response:
I have placed the empirical context at the beginning of the results and discussion (paragraph 4)
Important: the theoretical basis of the study should go deeper in some concepts. For example: it is unclear what it is meant with philosophy: is it the same than lifestyle (not really, so please clarify the use of this term). Please, make sure that your theoretical approach has some aspect of ORIGINALITY in comparison with the numerous studies about food and wine in rural tourism. I'm wondering whether the focus of this study is on the promotion of agriturismo, as this is what it is commented at the end of the paper. Please, clarify and go deeper in the theoretical approach/perspective.
Response:
I have changed the title of my paper in: Seeking gastronomic, healthy and social experiences in Tuscan agritourism facilities.
I hope it is ok
Pay attention to the order of the various topics within this section. As it is now, the text "jumps" from a topic to another and then back to the first topic and so on. A possible order for the Introduction might be: food & drink (importance, identity issues...), food & drink in rural tourism, presentation of ONE main RESEARCH QUESTION, + final paragraph presenting the structure and content of the following sections. Similarly, the theory chappter could begin with some considerations about already available studies about food tourism in rural areas, and then highlights what YOUR study will contribute with.
Response:
In a new version of the paper, I have ordered the section and I derived the questions researcher from the previous literature.
METHOD
The choice of Toscana as the emprical setting of the study should be explained.
More info about the sources of data, in particular the reviews: it seems that the reviews are from the agriturismo's webpage (not very reliable, as the agriturirsmo might delete bad reviews), or other online platforms? This is an aspect that should be explained. The possibility of faked reviews should also be commented. A table including such ino, as well as the number of reviews could be useful.
Response:
The data were extrapoled from the websites of the Tuscan agritourism facilities. For the visitors’ evaluation of the all structures characteristics and for the attributes of restaurant were considered both the positive and the negative scores (from 0 to 10). In the qualitative analysis, instead, in order to consider the gastronomic, healthy lifestyle and social experiences in agritourism facilities were extrapolated only the positive expressions that the visitors used to describe these experiences.
Please insert a CONCLUSION section that summarizes the MAIN findings, and highlights the theoretical and practical contribution of your study.
Avoid very short paragraphs (1 sentence)
Response:
I have inserted a Conclusion section (5) with a summary of the main finding and I have showed the theoretical and practical contribution of my study. Furthermore, I have avoided short sentences.
Thank you a lot for your suggestions
Best regards
Reviewer 2 Report
The paper evaluates the tourists regarding their consuming of local and traditional food and drink when they spent their vacations in the Tuscany. The research presents some critical points:
1) in the abstract is indicated the sample size (1,886) but in the paper this number is never been citated;
2) it is not clear from the abstract which method of analysis will be used;
3) the definition of food tourist (from row 56 to row 58) not has references;
4) the research questions are not sufficiently justified;
5) the Paragraph 2 is really scarce!
6) in the Paragraph 3 the results of the survey do not provide any innovative elements;
7) the author do not explain which methodology is been utilized to develop the qualitative analysis. Textual analysis????
Author Response
Comments of Reviewer 2:
The paper evaluates the tourists regarding their consuming of local and traditional food and drink when they spent their vacations in the Tuscany.
The research presents some critical points:
1) in the abstract is indicated the sample size (1,886) but in the paper this number is never been citated;
Response:
See line 250 of new version of the paper and Table 3 - Frequently occurring positive comments by visitors (line 355).
2) it is not clear from the abstract which method of analysis will be used;
Response:
See line 19 of the abstract (new version)
3) the definition of food tourist (from row 56 to row 58) not has references;
Response:
See lines 35-37 of the Introduction (new version of the paper). A ‘food tourist’ may be defined as a person who selects a travel destination as a result of the food experiences it can offer. Therefore, such a person gives great importance to the food, meal preparation and food-related activities offered at a destination.
4) the research questions are not sufficiently justified;
Response:
In the new version the reserch questions were derived from the Theoretical background and literature review. See from 72 to 170 line of the new version of the paper
5) the Paragraph 2 is really scarce!
Response:
The paragraph 2, in the new version of the paper, has been enriched with others previous studies on the topic. The reserch questions were derived mostly from these studies
6) in the Paragraph 3 the results of the survey do not provide any innovative elements;
Response:
In the new version the paragraph 3 has been rewritten and now I think that provides some innovative elements (see from 233 to 386 line).
7) the author do not explain which methodology is been utilized to develop the qualitative analysis. Textual analysis?
Response:
The Textual analysis, see lines 182 and183
Thank you a lot for your comments
Best regards
Reviewer 3 Report
This is a very interesting study, that provides interesting results. The sample data is significative and the application to the agritourism context is quite relevant. However, the research does not present a novelty to the state of the art and the authors should better provide the reader with the information of the gap and contribution. In fact, the authors might write a more extent literature review, contextualizing the use of digital platforms in the choice of local food and drinking, through the user generated contents, and explain what are the topics most shared, in the literature description. Alter that more focused literature review, the authors will be in a better position to propose a research on the specific context of agritourism, what may extent the knowledge in the field.
I suggest the following readings enrich the literature review:
Akdag, G., Guler, O., Dalgic, A., Benli, S. and Cakici, A. (2018), "Do tourists’ gastronomic experiences differ within the same geographical region? A comparative study of two Mediterranean destinations: Turkey and Spain", British Food Journal, Vol. 120 No. 1, pp. 158-171.
Cafiero, C., Palladino, M., Marcianò, C. and Romeo, G. (2019), "Traditional agri-food products as a leverage to motivate tourists: A meta-analysis of tourism-information websites", Journal of Place Management and Development
Garibaldi, R. and Pozzi, A. (2018), "Creating tourism experiences combining food and culture: an analysis among Italian producers", Tourism Review, Vol. 73 No. 2, pp. 230-241
Król, K. (2019), "Forgotten agritourism: abandoned websites in the promotion of rural tourism in Poland", Journal of Hospitality and Tourism Technology, Vol. 10 No. 3, pp. 431-442.
Leong, Q., Ab Karim, S., Awang, K. and Abu Bakar, A. (2017), "An integrated structural model of gastronomy tourists’ behaviour", International Journal of Culture, Tourism and Hospitality Research, Vol. 11 No. 4, pp. 573-592.
Lim, X., Ng, S., Chuah, F., Cham, T. and Rozali, A. (2019), "I see, and I hunt: The link between gastronomy online reviews, involvement and behavioural intention towards ethnic food", British Food Journal.
Mohamed, M., Hewedi, M., Lehto, X. and Maayouf, M. (2019), "Marketing local food and cuisine culture online: a case study of DMO’s websites in Egypt", International Journal of Tourism Cities
Oliveira, B. and Casais, B. (2019), "The importance of user-generated photos in restaurant selection", Journal of Hospitality and Tourism Technology, Vol. 10 No. 1, pp. 2-14.
Pérez Gálvez, J., Torres-Naranjo, M., Lopez-Guzman, T. and Carvache Franco, M. (2017), "Tourism demand of a WHS destination: an analysis from the viewpoint of gastronomy", International Journal of Tourism Cities, Vol. 3 No. 1, pp. 1-16.
Sanchez-Cañizares, S. and Castillo-Canalejo, A. (2015), "A comparative study of tourist attitudes towards culinary tourism in Spain and Slovenia", British Food Journal, Vol. 117 No. 9, pp. 2387-2411.
Anaya-Sánchez, R., Molinillo, S., Aguilar-Illescas, R. and Liébana-Cabanillas, F. (2019), "Improving travellers' trust in restaurant review sites", Tourism Review, Vol. 74 No. 4, pp. 830-840.
Anyway, the gap has to be clarified after this literature review and a better explanation of why agritourism is relevant to research, considering what is already known regarding gastronomy tourism and e-wom e in restaurants’ digital platforms.
The authors write: As Morgan (2010) claims: “Food is … vital 25 to human health and well-being in a way that the products of other industries are not, and this remains the 26 quintessential reason as to why we attach such profound significance to it “. (Morgan 2010).
You don’t need to cite the same sentence before and in the end. And the reference should appear before the dot. Further, when a direct citation is used, the number of the page of the original article has to appear.
Pg 1 line 26 Food plays…
The flow of the paragraphs are disconnected. There are paragraphs that continue the same idea of the previous paragraph and might be joined.
The paper requires an English revision.
The introduction informs the research questions but does not proves how the RQ emerges, I mean why they are gaps of knowledge. The authors have to say what is known util now so that the RQ make sense as a solution for the gap of knowledge. After the introduction, a literature review is required to address the state of the art of consumer expectations regarding gastronomy tourism and the user generated contents in digital platforms about restaurants.
In the methodology, how the variables were chosen? From the literature review? May you provide more information about that?
I also suggest a better discussion of results, reflecting about the particularities of results in the agritourism context comparing with previous knowledge in general gastronomy tourism.
Add a conclusion and a section with limitations and Future research.
Good Luck with your research.
Author Response
Reviewer 3
This is a very interesting study, that provides interesting results. The sample data is significative and the application to the agritourism context is quite relevant. However, the research does not present a novelty to the state of the art and the authors should better provide the reader with the information of the gap and contribution. In fact, the authors might write a more extent literature review, contextualizing the use of digital platforms in the choice of local food and drinking, through the user generated contents, and explain what are the topics most shared, in the literature description. Alter that more focused literature review, the authors will be in a better position to propose a research on the specific context of agritourism, what may extent the knowledge in the field.
I suggest the following readings enrich the literature review:
Response:
In a new version of the paper, I have enriched the literature and I have inserted all the following references that you have suggested. Thank you a lot for your important indications.
Akdag, G., Guler, O., Dalgic, A., Benli, S. and Cakici, A. (2018), "Do tourists’ gastronomic experiences differ within the same geographical region? A comparative study of two Mediterranean destinations: Turkey and Spain", British Food Journal, Vol. 120 No. 1, pp. 158-171.
Cafiero, C., Palladino, M., Marcianò, C. and Romeo, G. (2019), "Traditional agri-food products as a leverage to motivate tourists: A meta-analysis of tourism-information websites", Journal of Place Management and Development
Garibaldi, R. and Pozzi, A. (2018), "Creating tourism experiences combining food and culture: an analysis among Italian producers", Tourism Review, Vol. 73 No. 2, pp. 230-241
Król, K. (2019), "Forgotten agritourism: abandoned websites in the promotion of rural tourism in Poland", Journal of Hospitality and Tourism Technology, Vol. 10 No. 3, pp. 431-442.
Leong, Q., Ab Karim, S., Awang, K. and Abu Bakar, A. (2017), "An integrated structural model of gastronomy tourists’ behaviour", International Journal of Culture, Tourism and Hospitality Research, Vol. 11 No. 4, pp. 573-592.
Lim, X., Ng, S., Chuah, F., Cham, T. and Rozali, A. (2019), "I see, and I hunt: The link between gastronomy online reviews, involvement and behavioural intention towards ethnic food", British Food Journal.
Mohamed, M., Hewedi, M., Lehto, X. and Maayouf, M. (2019), "Marketing local food and cuisine culture online: a case study of DMO’s websites in Egypt", International Journal of Tourism Cities
Oliveira, B. and Casais, B. (2019), "The importance of user-generated photos in restaurant selection", Journal of Hospitality and Tourism Technology, Vol. 10 No. 1, pp. 2-14.
Pérez Gálvez, J., Torres-Naranjo, M., Lopez-Guzman, T. and Carvache Franco, M. (2017), "Tourism demand of a WHS destination: an analysis from the viewpoint of gastronomy", International Journal of Tourism Cities, Vol. 3 No. 1, pp. 1-16.
Sanchez-Cañizares, S. and Castillo-Canalejo, A. (2015), "A comparative study of tourist attitudes towards culinary tourism in Spain and Slovenia", British Food Journal, Vol. 117 No. 9, pp. 2387-2411.
Anaya-Sánchez, R., Molinillo, S., Aguilar-Illescas, R. and Liébana-Cabanillas, F. (2019), "Improving travellers' trust in restaurant review sites", Tourism Review, Vol. 74 No. 4, pp. 830-840.
Anyway, the gap has to be clarified after this literature review and a better explanation of why agritourism is relevant to research, considering what is already known regarding gastronomy tourism and e-wom e in restaurants’ digital platforms.
Response:
I have explained in a better way why the agritourism is relevant to research in relation to what is already found in others previous studies.
The authors write: As Morgan (2010) claims: “Food is … vital 25 to human health and well-being in a way that the products of other industries are not, and this remains the 26 quintessential reason as to why we attach such profound significance to it “. (Morgan 2010).
You don’t need to cite the same sentence before and in the end. And the reference should appear before the dot. Further, when a direct citation is used, the number of the page of the original article has to appear.
Pg 1 line 26 Food plays…
The flow of the paragraphs are disconnected. There are paragraphs that continue the same idea of the previous paragraph and might be joined.
The paper requires an English revision.
The introduction informs the research questions but does not proves how the RQ emerges, I mean why they are gaps of knowledge. The authors have to say what is known util now so that the RQ make sense as a solution for the gap of knowledge. After the introduction, a literature review is required to address the state of the art of consumer expectations regarding gastronomy tourism and the user generated contents in digital platforms about restaurants.
Response:
In the new version of the paper, I have proposed a shorter Introduction (paragraph 1) Theoretical background and literature review, instead, are reported in the paragraph 2 (see the new version of the paper from pag. 2 to pag 4).
In the methodology, how the variables were chosen? From the literature review? May you provide more information about that?
Response:
The variables have been chosen on the base of the literature review in relation to those available on the websites of the agritourim facilities associate with Agritourismo.it organization. The aim is to provide some indication on offer side (altitude, visitors frequency) and more information on demand side (visitors’ evaluations, typologies of visitors, motivational factors that drive the visitors to choice agritorurism facility as a destination of their short or long vacations.
I also suggest a better discussion of results, reflecting about the particularities of results in the agritourism context comparing with previous knowledge in general gastronomy tourism.
Response:
I have discussed in more details the findings of the study in relation to previous literature
Add a conclusion and a section with limitations and Future research.
Response:
I have added a conclusion paragraph that contain summery considerations, study’s limitation and some suggesting for future research
Thank you a lot for your suggestions
Best regards
Reviewer 4 Report
Thank you for bringing the interesting topic to the journal. Here are several points that the authors should improve:
(1) In the section of [2. Materials and Methods], there are not enough information about all required components: (a) what are the variables that the authors measure? (b) how to measure them? (c) how to operationalize them? (d) why Tobit is chosen for the best analytic tool instead of other analytics.
(2) In the section of [2. Materials and Methods], the function seems to be more likely "black box." The authors are recommend to express the function more clearly. For instance, it should be "EVALREST i,t = α + βjΣ(Attributes of agritourism facilities) i,t + βmΣ(Typology of visitors) i,t + βnΣ(Motivational factors) i,t + εi,t." Also, there should be a specific list of attributes of agritourism facilities, typology of visitors, motivational factors. Without these information, all explanation in the manuscript is just "black box."
(3) As indicated in (2), the authors should explain the 33 factors in the model with answering at least these questions; (a) what are they? (b) why are they chosen? (c) are there any relevant conceptual framwork or theory with the variable list?
(4) Research questions were suddenly shown without any theoretical background or conceptual framework. Is the paper an exploratory study? If so, the authors should explain the study is an explanatory and the reason why it should be and exploratory study.
(5) In the research question 1, a term "philosophy of life" suddenly showed up. What is the meaning of philosophy of life? The authors should explain the concept of the word at least one time prior to the research question.
(6) Research methodology is not sufficiently explained. Based on the four research question, Tobit is not only the way to analyze. Also, in the result section, there is another analysis (e.g., descriptive analysis instead of Tobit) used for Research question 1 and 3. The authors should explain sufficiently to match each research question with the appropriate analytic method. For instance, for the research question 1, the authors should explain that they used the descriptive analysis for the research question 1; also, they should justify the reason why they use descriptive analysis in the first research question. Samely, in the second and third research question, the authors should explain why they utilize the different analytic method per each question and justify them.
(7) For the research question 4, the answer was shown in discussion separately. In this case, then the research question 4 is not really research question. It should be implication based on the previous three research question.
(8) There are many paragraphs with single-sentence. Highly recommend to reorganize those single-sentence paragraph into the appropriate form of structure.
Author Response
Reviewer 4
Thank you for bringing the interesting topic to the journal. Here are several points that the authors should improve:
(1) In the section of [2. Materials and Methods], there are not enough information about all required components: (a) what are the variables that the authors measure? (b) how to measure them? (c) how to operationalize them? (d) why Tobit is chosen for the best analytic tool instead of other analytics.
(2) In the section of [2. Materials and Methods], the function seems to be more likely "black box." The authors are recommend to express the function more clearly. For instance, it should be "EVALREST i,t = α + βjΣ(Attributes of agritourism facilities) i,t + βmΣ(Typology of visitors) i,t + βnΣ(Motivational factors) i,t + εi,t." Also, there should be a specific list of attributes of agritourism facilities, typology of visitors, motivational factors. Without these information, all explanation in the manuscript is just "black box."
(3) As indicated in (2), the authors should explain the 33 factors in the model with answering at least these questions; (a) what are they? (b) why are they chosen? (c) are there any relevant conceptual framwork or theory with the variable list?
Response:
In the second step, the Tobit regression analysis was adopted to estimate how much of the variation in the visitors’ evaluation of restaurant is explained by the independent variables. In this model, devised by Tobin (1958), was possible uses, a difference of others alternative techniques, all the variables, both those at limiting values (usually 0) and those above it to estimate a regression line.
The dependent variable: Evaluation of restaurant (EVALREST) is a discrete variable, with a variation range between 0 and 10; α = the intercept of the regression equation; βk = coefficients of independent variables, where k = 1, 2, 3,…33, ε = error term.
It is hypothesized that this evaluation is influenced by the three following blocks of the independent variables:
Attributes of agritourism facilities:
Altitude = location of the agritourism facility meters above sea level. Discrete variable,
Evaluation (score) of all structures’ characteristics = Discrete variable with a variation range between 0 and 10. The hypothesis is that the visitors’ evaluation of the gastronomic experience is positively related to all agritourism facilities attributes (countryside, activities, facilities, traditions, cultural heritage and so on).
Visitor frequency (Nr/Ny) = The variable is aspect to have positive relationship with the dependent variable because it is hypothesized that a high number of reviews posted by visitors that partaking their experience online is a positive sign for the reputation of foodservices.
Typology of visitors
It is hypnotizing that some typologies of visitors are more interested in this type of the vacation and influenced in manner more positive the dependent variable. Furthermore, their judges depend on their level of the satisfaction with these structures.
Motivational factors
The motivational factors that have driven the visitors to choice this type of vacation can influence in different way their gastronomic, healthy lifestyle and social experiences in the agritourism structures.
The following is the testable model for Tobit regression:
EVALREST i,t = α + βjΣ(Attributes of agritourism facilities)i,t + βmΣ(Typology of visitors) i,t + βnΣ (Motivational factors) i,t + εi,t
(4) Research questions were suddenly shown without any theoretical background or conceptual framework. Is the paper an exploratory study? If so, the authors should explain the study is an explanatory and the reason why it should be and exploratory study.
Response:
In the new version of the paper, I have proposed a shorter Introduction (paragraph 1) Theoretical background and literature review, instead, are reported in the paragraph 2 (see the new version of the paper from pag. 2 to pag 4).
In a new version of the paper, I have ordered the section and I derived the questions researcher from the previous literature.
(5) In the research question 1, a term "philosophy of life" suddenly showed up. What is the meaning of philosophy of life? The authors should explain the concept of the word at least one time prior to the research question.
Response:
I have changed the title of my paper in: Seeking gastronomic, healthy and social experiences in Tuscan agritourism facilities.
I hope it is ok
(6) Research methodology is not sufficiently explained. Based on the four research question, Tobit is not only the way to analyze. Also, in the result section, there is another analysis (e.g., descriptive analysis instead of Tobit) used for Research question 1 and 3. The authors should explain sufficiently to match each research question with the appropriate analytic method. For instance, for the research question 1, the authors should explain that they used the descriptive analysis for the research question 1; also, they should justify the reason why they use descriptive analysis in the first research question. Samely, in the second and third research question, the authors should explain why they utilize the different analytic method per each question and justify them.
(7) For the research question 4, the answer was shown in discussion separately. In this case, then the research question 4 is not really research question. It should be implication based on the previous three research question.
(8) There are many paragraphs with single-sentence. Highly recommend to reorganize those single-sentence paragraph into the appropriate form of structure.
Response:
See a new version of the paper where I have derived research questions from the previous literature and where I have matched each research question with appropriate analytic method
Thank you a lot for your suggestions
Best regards
Round 2
Reviewer 1 Report
The manuscript quality has improved.
In my opinion, the study contribution is now better highlighted and the paper is useful and enjoyable for the readers interested in food tourism.
(please check MINOR mistake line 107)
Reviewer 2 Report
The paper has improved considerably compared to the previous version.
The paper has improved considerably compared to the previous version. All my remarks have been properly integrated into the text.
Reviewer 3 Report
The authors followed all my suggestions and i consider the paper can be published.
Reviewer 4 Report
I reviewed the whole paper; and the authors improved a lot of the manuscript. Specifically, the explanation about conceptual framework and theoretical background is more understandable than the original manuscript. If the authors describe the conceptual framework with better format (e.g., figure), it would be better for readers to understand. However, the current format is still good. Thank you for giving me an opportunity to read such an interesting manuscript.